# Segmenting functional tissue units across human organs using community-driven development of generalizable machine learning algorithms

Yashvardhan Jain [1] ✉, Leah L. Godwin[1], Sripad Joshi [1], Shriya Mandarapu [1], Trang Le[2,3], Cecilia Lindskog [4], Emma Lundberg [2,3,5,6] & Katy Börner [1] ✉

The development of a reference atlas of the healthy human body requires automated image segmentation of major anatomical structures across multiple organs based on spatial bioimages generated from various sources with differences in sample preparation. We present the setup and results of the Hacking the Human Body machine learning algorithm development competition hosted by the Human Biomolecular Atlas (HuBMAP) and the Human Protein Atlas (HPA) teams on the Kaggle platform. We create a dataset containing 880 histology images with 12,901 segmented structures, engaging 1175 teams from 78 countries in community-driven, open-science development of machine learning models. Tissue variations in the dataset pose a major challenge to the teams which they overcome by using color normalization techniques and combining vision transformers with convolutional models. The best model will be productized in the HuBMAP portal to process tissue image datasets at scale in support of Human Reference Atlas construction.

Constructing the Human Reference Atlas (HRA) requires harmonization and analysis of massive amounts of imaging and other data to capture the organization and function of major anatomical structures and cell types[1–3]. A key task is the segmentation of major anatomical structures—from the whole body to the single-cell level. Functional tissue units (FTUs) help bridge the scale difference and are used as a stepping stone from the organ to the single-cell level. FTUs are defined as the smallest tissue organization that performs a unique physiologic function and is replicated multiple times in a whole organ[4]. The spatial organization of FTUs matters and strongly impacts the function of an organ. FTUs that are diseased have different cell type populations and possibly different sizes and shapes, or are altered in the number or organization of FTUs within an organ. Several organ atlas efforts within

the HuBMAP[1] effort are now focusing on cell types, cell states, and biomarkers in specific FTUs[5,6]. Being able to segment FTUs is an important part of identifying cell types and their gene/protein expression patterns within an FTU.

To segment anatomical structures in histological tissue sections efficiently, human intelligence must be combined with machine intelligence to overcome several challenges: segmenting histological images manually is labor-intensive, there are challenges with inter-observer variability, and there might be subtle differences and details that cannot be recognized or may be missed by the human eye. In support of efficient and high-quality tissue segmentation, human-in-the-loop approaches have been implemented[7,8]. Here, human expertise is used to identify and prepare relevant image data; design,

[1]Department of Intelligent Systems Engineering, Luddy School of Informatics, Computing, and Engineering, Indiana University, Bloomington, IN 47408, USA. [2]Science for Life Laboratory, School of Engineering Sciences in Chemistry, Biotechnology and Health, KTH - Royal Institute of Technology, Stockholm, Sweden. [3]Department of Bioengineering, Stanford University, Stanford, CA 94305, USA. [4]Department of Immunology, Genetics and Pathology, Division of Cancer Precision Medicine, Uppsala University, Uppsala, Sweden. [5]Department of Pathology, Stanford University, Stanford, CA 94305, USA. [6]Chan Zuckerberg Biohub, San Francisco, CA 94305, USA. ✉e-mail: yashjain@iu.edu; katy@indiana.edu

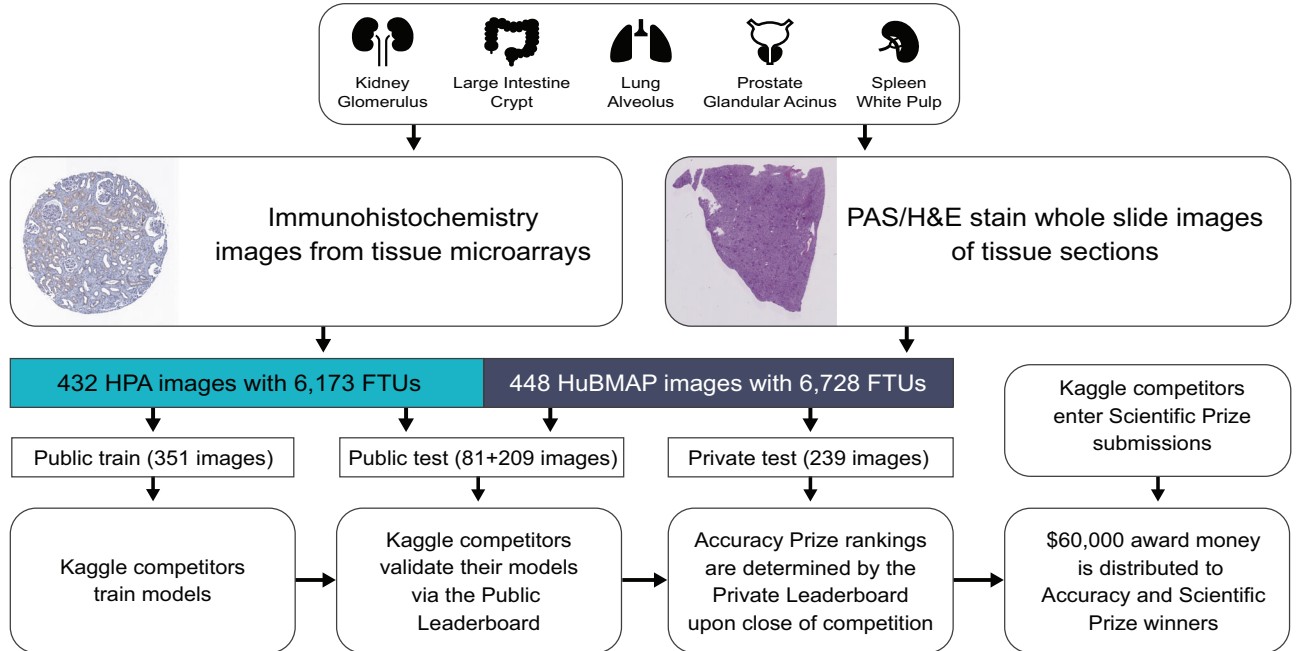

**Fig. 1 | Overview of competition setup.** Tissue data for five organs collected within HPA and HuBMAP using different tissue harvesting and processing protocols are collected and divided into a public training, public test, and private test dataset. Kaggle teams use the public training data to train their models; they then submit the models to the Kaggle Submission Portal to receive performance scores computed using the public test data. At the end of the competition, when all teams submit their best algorithm solutions, their score on the private test set determines the Performance Prize winners.

optimize, train, and run effective machine learning (ML) algorithms; and interpret results. Once high-quality ML training datasets are compiled, generation, and federation pipelines are set up, ML algorithms can be trained and optimized to segment image data at scale. As new datasets are segmented, and these ML segmentations are validated and/or improved by human experts, ML algorithm performance can be further improved using this additional training data. Over the last decade, much work has been done on segmenting histological images; most of this work focuses on single-cell segmentation[8] or target structures in a single organ[7,9–11], including functional tissue units (FTUs). Although there are foundation models[12] for image segmentation emerging, their performance on medical imaging tasks is fairly limited[13]. To the best of our knowledge, there exist no ML algorithms that can segment FTUs across multiple organs in datasets from different laboratories.

In 2021, the Human BioMolecular Atlas Program (HuBMAP) conducted a Kaggle competition[4,14] that focused exclusively on the segmentation of renal glomeruli in PAS-stained histological images of kidney tissue, engaging 1200 teams from 60 countries. The winning model from this competition was validated and productized in the HuBMAP data portal to run on all PAS-stained kidney tissue data at scale. In parallel, the Human Protein Atlas (HPA)[2] conducted two Kaggle competitions[15–18] that focused on classification of subcellular patterns in cultivated cells in microscope confocal images, engaging nearly 3000 teams across the two competitions. In addition to the confocal images of cultivated cells, the HPA has also generated >10 million immunohistochemically stained images from 44 major tissue types of the human body, corresponding to all the major normal organs[19].

HuBMAP and HPA partnered to address two major challenges when constructing the Human Reference Atlas (HRA): (1) standardization of data coming from various sources (different sample preparation and staining protocol, different equipment readout, etc.) and (2) robust and generalized segmentation of FTUs across various tissue types. The two teams hosted a joint competition on Google's ML community platform, Kaggle[20], inviting competitors to develop

machine learning algorithms that correctly segment FTUs of different shapes and sizes across five organs. This paper details the competition design (see Fig. 1) and highlights the major challenges of the competition and the strategies used by the winning teams. We present an analysis of model failures for each FTU type and the impact of additional metrics on team performance and competition rankings. In addition, we analyze and visualize competition dynamics and code performance improvements by 1175 teams making 39,568 code submissions over the 3-months period.

## Results

### Competition design and performance criteria

The aim of the Hacking the Human Body[21] competition was to develop machine learning algorithms for the segmentation of functional tissue units in five human organs using histology images sourced from two different consortia, namely HuBMAP and HPA (see Fig. 2). The competition was designed to build algorithms that are generalizable across multiple organs and robust across dataset differences such as image resolution differences, color differences, artifacts, staining differences, etc. HPA's primary interest in this competition is that models that can segment FTUs in tissue sections can pave the way for more quantitative analysis of the data generated for the Tissue Atlas section of the HPA, e.g., to understand differences in protein expression patterns within FTUs as donor sex, ethnicity, or age change, or comparison of expression patterns of different proteins in the same donors. Human Reference Atlas construction in HuBMAP and other consortia use FTUs to characterize local cell neighborhoods with well-defined physiologic functions; they are interested in capturing differences in FTU numbers, sizes, and shapes for different donor demographics in health and disease. Being able to segment FTUs in tissue sections in histology images is an important step for characterizing their morphology, cell types and gene/protein expression patterns.

The HPA and HuBMAP datasets cover five FTUs in five organs, namely renal glomeruli in the kidney (renal corpuscle, UBERON:0001229), crypts in the large intestine (crypt of Lieberkühn of the large intestine, UBERON:0001984), alveoli in the lung

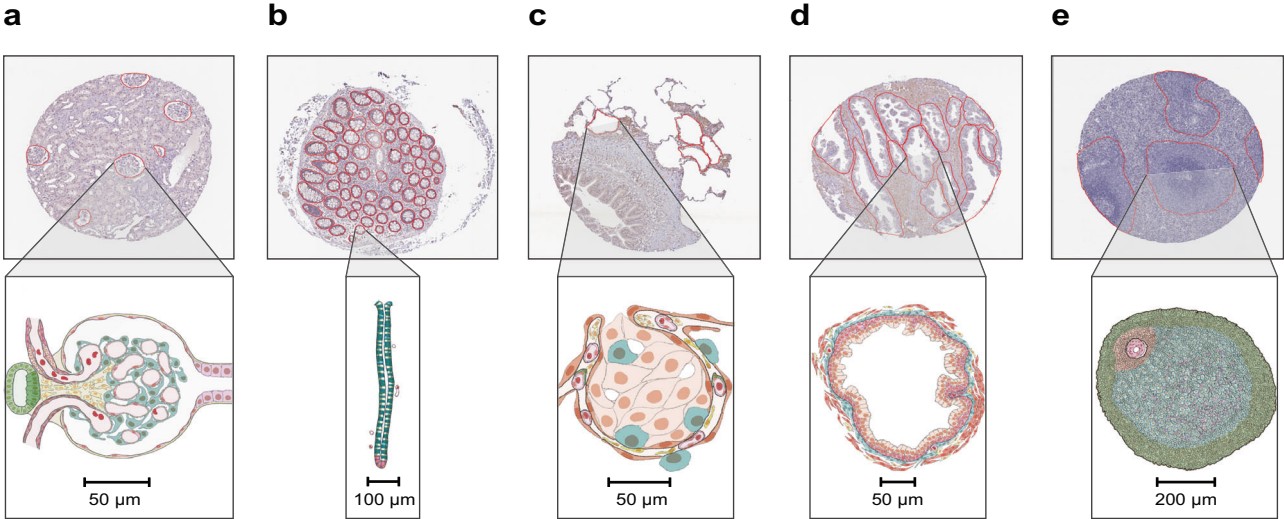

**Fig. 2 | Exemplary tissue microarray cores with FTU segmentations outlined in red and illustrations for all five FTUs. a** Glomerulus in the kidney. **b** Crypt in the large intestine (top: perpendicular cross-section, bottom: lengthwise cross-section). **c** Alveolus in the lung. **d** Glandular acinus in the prostate. **e** White pulp in the spleen.

(pulmonary alveolus, UBERON:0002299), glandular acini in the prostate (prostate glandular acinus, UBERON:0004179), and white pulps in the spleen (white pulp, UBERON:0001959). A dataset of 880 images was compiled, containing 432 images from HPA and 448 images from HuBMAP. This dataset was split into a training dataset of 351 images, and a private and public test dataset of 529 images (see Table 1 for detailed breakdown). The HuBMAP dataset was preprocessed into a set of smaller tiled images (see "Methods") to make HPA and HuBMAP datasets more comparable and to ensure teams could fully focus on developing machine learning algorithms rather than handling large format whole slide images (WSIs); providing image tiles also made the competition more equitable as computing requirements such as high RAM and high GPU access were not needed to develop code. Participants were allowed to use any external, publicly available data. All code submitted via the Kaggle submission portal was run on the public and private test sets, leading to team rankings on the public and private leaderboards, respectively (see "Methods"). The algorithm performance was measured using the mean Dice coefficient[22] on the test sets. The top-3 teams on the private leaderboard at the end of the competition won the performance prizes. In addition, teams submitted entries to win the scientific prizes and the diversity prize (see "Methods").

The major challenge in this competition was to build ML code solutions that are trained on one type of staining method (from HPA) and can generalize to cover other types of staining methods (from HuBMAP) during inference. Consequently, teams developed strategies to deal with differences in terms of resolution, color, tissue thickness, etc. (see details in Supplementary Notes 1–3). In addition, teams had to optimize code for multiple organs, as lower performance on any organ would negatively impact the overall score. Other challenges included small training set size, uneven train/test split, and class imbalance, which motivated teams to build optimal solutions to extract maximum signal from the training data.

## Performance and winning strategies

The winning team for the performance prize reached a mean Dice score of 0.835 on the private leaderboard, followed closely by the second (0.833) and third (0.832) place winners. The score drops by 0.005 for the fourth-place solution, reaching a mean Dice score of 0.827. The top-3 teams made a combined total of 619 submissions throughout the 3-months competition period. In general, the teams

found the kidney and large intestine to be the easiest classes, followed by the spleen, prostate, and lung. Lung was the most challenging class in the competition (see Table 2 for Dice scores per organ for the three winning teams), primarily due to the variations in alveoli segmentations as they contained both collapsed and uncollapsed alveoli, as well as the variations in cuts (elongated vs. circular, see Supplementary Note 4).

The main strategies that helped the teams to increase performance scores were data augmentation (geometric, color, distortions, scales) which involves artificially increasing the amount of data by adding many minor alterations to the original data, stain normalization (Vahadane method[23,24]), using external data for training, and pseudo labeling which involves adding increasingly confident label predictions from a semi-supervised training loop. All three winning teams used some version of all these strategies. Interestingly, the fourth-place solution only used heavy stain normalization (reducing the importance of color in a model and forcing it to look for other cues in the images) and no external data or pseudo labeling, and was able to reach a mean Dice score of 0.827. In addition, vision transformers proved to be more efficient compared to traditional convolutional networks due to their ability to capture long-range dependencies. However, such models are more sensitive to hyperparameter tuning and data changes. The teams found SegFormer[25] models to be the best-performing vision transformers. Since the SegFormer license is not completely open-source, teams also explored other vision transformer models and found Co-scale conv-attentional image Transformer (CoaT)[26] models to be an effective replacement which performed equally well, while Swin[27] transformers performed poorly. Finally, the second-place solution showed that using bio-relevant auxiliary tasks such as organ classification and pixel size prediction also helps boost performance.

The first and third[28] place winning teams also performed ablation studies (see Supplementary Tables 1 and 2) to assess the impact of different strategies on performance. The three most effective strategies were building ensembles of multiple models with at least one vision transformer model, using external data and pseudo labeling, and heavy data augmentation and/or stain normalization strategies. Team 3 used pixel size adaptation and histogram matching to boost performance. Team 2 found that heavy encoders and networks with larger input resolutions worked better. Team 1 showed that while ensembles provide the best performance, the SegFormer mit-b4 model[25] provides the best score (0.828) as a single model. This is an

**Table 1 | Metadata for the final public HPA, private HPA and HuBMAP data that comprised the competition dataset**

| | Number of images | Number of unique male/female donors | Donor age range | Number of FTUs |
|---|---|---|---|---|
| **Public HPA data** | | | | |
| Kidney | 99 | 5/3 | 28–73 | 337 |
| Large intestine | 58 | 3/4 | 47–84 | 3107 |
| Lung | 48 | 4/4 | 21–78 | 188 |
| Prostate | 93 | 8/0 | 37–76 | 1097 |
| Spleen | 53 | 4/4 | 21–82 | 167 |
| **Public HPA totals** | 351 | 24/15 | 21–84 | 4896 |
| **Private HPA data** | | | | |
| Kidney | 19 | 4/3 | 28–70 | 69 |
| Large intestine | 18 | 3/4 | 47–84 | 892 |
| Lung | 14 | 3/4 | 43–78 | 66 |
| Prostate | 18 | 7/0 | 37–76 | 212 |
| Spleen | 12 | 3/3 | 21–74 | 38 |
| **Private HPA totals** | 81 | 20/14 | 21–84 | 1277 |
| **HPA totals** | 432 | 29/17 | 21–84 | 6173 |
| **HuBMAP data** | | | | |
| Kidney | 79 | 8/7 | 20–77 | 538 |
| Large intestine | 43 | 2/2 | 22–48 | 1966 |
| Lung | 115 | 16/7 | 19–73 | 2630 |
| Prostate | 98 | 10/0 | 18–57 | 1202 |
| Spleen | 113 | 9/2 | 0–47 | 392 |
| **HuBMAP totals** | 448 | 45/18 | 0–77 | 6728 |
| **Totals** | 880 | 74/35 | 0–84 | 12,901 |

All donors in the private HPA dataset are present in the public HPA dataset. All donors in the HuBMAP dataset are different from donors in the HPA dataset.

**Table 2 | Mean Dice scores per organ for top-3 teams based on the private test set**

| Team | Kidney | Intestine | Lung | Prostate | Spleen | Overall |
|---|---|---|---|---|---|---|
| Team 1 | 0.96401 | 0.89676 | 0.72664 | 0.85004 | 0.83862 | 0.83562 |
| Team 2 | 0.9665 | 0.88931 | 0.72092 | 0.84851 | 0.84157 | 0.83393 |
| Team 3 | 0.9491 | 0.86232 | 0.73599 | 0.84806 | 0.84211 | 0.83266 |

The performance of all three teams is comparable for each organ. All teams have the lowest scores on the lung images, and the highest scores on the kidney images.

important result as ensembles are resource intensive and can be impractical for processing images at scale. A single model combined with carefully selected image preprocessing strategies can be a good choice in production environments. An extended summary of the three winning solutions can be found in Supplementary Notes 1–3. All code implementations and datasets are publicly available on GitHub and Zenodo[29–31] (see "Data availability" and "Code availability").

## Qualitative analysis of predictions

To assess the strengths and failures of the winning models, predicted segmentation masks are compared with the ground truth masks to visualize per-pixel false positives and false negatives for five best and five worst cases (per organ; based on the Dice score for the image). The images with the best Dice scores show that most of the disagreement between the predictions and the ground truth happens at the boundaries of the FTUs, but all models are generally able to predict at least some portion of all instances of the FTUs in the image. Supplementary Figs. 3–32 show the visualizations for all these cases.

The images with the worst Dice scores for each organ show similar trends of failure across all three models. For the kidney, sometimes the algorithm predicts sclerotic (diseased/unhealthy) glomeruli which were not included in the ground truth. In some cases, team 2 and team 3 (but not team 1) may predict large venous structures as well. For the large intestine, the models often under-segment rather than over-segment. The models have difficulty identifying the FTUs in the large intestine when the tissue section cuts through only the epithelial cells of the intestinal gland but not through the luminal space (see second image from top in Supplementary Fig. 6). The models struggle the most at defining the FTU boundaries in the spleen data (see predictions in Supplementary Figs.). The mantle zone between the white pulp and the red pulp is the most prone to prediction error, especially in

prediction with the lowest Dice score, possibly due to the decreased lymphatic cell concentration and subsequent reduction of staining difference. For the tubuloacinar prostate gland, the models trend towards over-segmenting as they also predict the glandular tubules of the prostatic gland while the ground truth only contains the glandular acini as the FTU of interest. The models seem to segment the entire gland, not just the acinus, leading to lower Dice scores. However, the models rarely predict non-glandular tissue in the prostatic gland, which indicates there is accurate discernment of functional vs non-functional tissue. The models' predictions in the lung tissue often miss alveoli that are not closed, i.e., alveoli that have had damage or rupture during the tissue sectioning process. Additionally, the ground truth labels for the lung have the noisiest labels as there are cases where alveolar structures are missing in the ground truth but are correctly predicted by the models. The worst-case prediction for team 2 incorrectly predicts almost the entire lung tissue image as alveolar structures, which hurts its score, but perhaps is an anomaly in prediction (see topmost image in Supplementary Fig. 18).

To further assess performance, the Intersection-Over-Union (IOU), also known as Jaccard Index[32,33], was calculated per image. Ranking the competition based on mean IOU, instead of mean Dice, changes the top-50 rankings to some extent, but the top-3 teams rank the same with a mean IOU of 0.7384, 0.7362, and 0.7333, respectively (see "Statistical analysis" in "Methods"). In addition, while boundary-based metrics like Hausdorff Distance[34] and the Hausdorff Distance at 95th percentile (HD95)[35] may help in further comparison between teams, these are not evaluated as they are not as relevant nor appropriate due to the presence of multiple structures per image as well as the presence of touching FTU structures[36].

Figure 3 shows the violin plots for mean Dice scores and mean IOU scores for all three teams, broken down by organs. For each violin plot, the individual image scores are also plotted as a swarm plot overlaid on top of the violin plots to show the spread and outliers.

## Participation analysis using meta kaggle

The competition ran from June 22, 2022 to September 22, 2022 and involved 1517 individual competitors from 78 countries that collaborated in 1175 teams. For 286 competitors, it was their first time participating in a Kaggle competition and 36 of them made the top-100 list on the private leaderboard during the competition run. In total, the teams made 39,568 submissions and created 922 public code notebooks. In addition, the participants created 224 public discussion forum posts and made 943 discussion comments. These metrics help understand the truly collaborative and globally inclusive nature of Kaggle competitions where teams interact extensively to share code, data, and knowledge.

Kaggle ranks all its users in five performance tiers based on their achievements and engagement on the platform, using their user Progression System[37]. In this competition, we had 22 Grand Masters, 103 Masters, 372 Experts, 559 Contributors, and 450 Novices participating (performance tier data is missing in Meta Kaggle for 11 users). The top-2 winning teams included experienced software engineers with a passion for machine learning and computer vision. The team winning third place consisted of computer scientists, machine learning researchers, and analysts. Many participants had biomedical

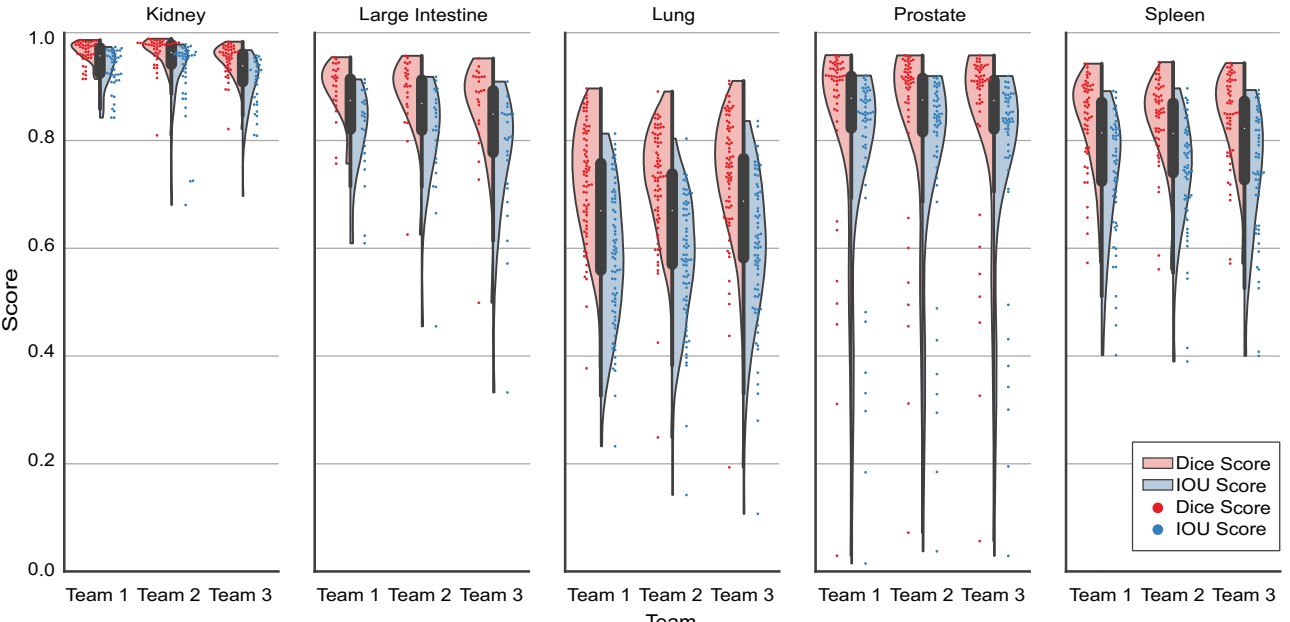

**Fig. 3 | Violin plots for top three teams per organ.** Distribution of mean Dice and IOU score per image (private test set) is shown as violin plots for each organ for all three winning teams, coded by Dice score (light red) and IOU score (light blue). A swarm plot is overlaid on the violin plots to show individual scores for each image, coded by Dice score (red) and IOU score (blue). Source data are provided as a source data file on Zenodo (see "Data availability").

backgrounds and shared their domain expertise generously via the discussion forums.

Figure 4 graphs the dynamics of the three-month competition. Figure 4a shows the number of teams and messages, and the progression of top leaderboard scores over the competition period. Note the sudden increase in the number of messages after the team merger deadline and winner announcement. The scores reached nearly 0.80 midway through the competition, after which improvements were made through fine-tuning solutions using techniques such as pseudo labels, using ensembles of multiple models, etc. Importantly, the public and private leaderboard scores remained similar throughout the competition leading to minimal change in rankings (also called shake-up) at competition end and indicating a good dataset split between public test and private test datasets. Figure 4b plots the number of submissions versus the highest private score; many of the 1175 teams have few submissions with low scores; some teams have many submissions with high scores. Team 1 submitted 264 times, team 2 submitted 100 times, and team 3 made 255 submissions.

### Scientific and diversity prizes

A total of six teams submitted their entries for the Scientific and Diversity prizes using a Google Form. The ten judges reviewed all submissions and graded them based on the rubric, ranking all submissions. Submission 5 and Submission 6 received the most points from all the judges for the two scientific prizes. Submission 5 focused on showcasing differences between a convolution model and a vision transformer model, the latter achieving better performance as their bigger receptive field helps analyze images in a global context which is more suitable for medical image segmentation tasks. In addition, it also showcased the importance of stain normalization in bridging the domain difference between the HPA and HuBMAP data. Submission 6 showcased the impact of noisy labels in the ground truth for training data and proposed a method to dynamically relabel missing annotations and minimize the gap between noisy and clean labels, thereby boosting performance by 4% on the private leaderboard.

All judges unanimously agreed Submission 1 should receive the Diversity and Presentation prize for building a team of diverse members and presenting their experiments and results in a well-documented and accessible manner.

## Discussion

Building the Human Reference Atlas is a challenging task that requires close collaboration by experts from different scientific domains to solve key data integration, modeling, and visualization challenges across spatial and temporal scales. Kaggle's open-source and community-driven nature makes it possible to bring in experts from industry, academia, and government; to try out algorithms that were originally developed for different application domains; and to discuss solutions and results publicly empowering many to develop innovative solutions. All data and code are shared openly as a benchmark for use in future algorithm performance exercises and comparisons. Kaggle and other code competition platforms make it possible to share the burden of effective data preprocessing; run and compare thousands of ML algorithms in a very short period of time; and build on and advance these solutions collectively; something that is not possible at this speed and scale if research is performed in individual labs.

The Hacking the Human Body competition showcased the value of vision transformers in biological image processing, with all three winning teams building model ensembles consisting of some or all vision transformer models. This is in stark contrast to the previous HuBMAP competition[14] (concluded in May 2021), where all winning teams used convolutional models, evincing the quick rise of transformer models in the field.

Sourcing ground truth labels for supervised learning tasks, especially in biomedical domains, is a time-consuming and expensive challenge. The participants used diverse techniques to overcome this challenge, including using additional unlabeled data and creating pseudolabels for training iteratively to improve performance using a semi-supervised approach. This, in addition to clever data augmentation and normalization techniques, turned out to be the key to building generalized solutions that can be deployed at scale.

While this competition provides innovative and high-performing solutions, there exist several known limitations: (1) since the models are trained on a small dataset, there is risk of model overfitting; (2) the

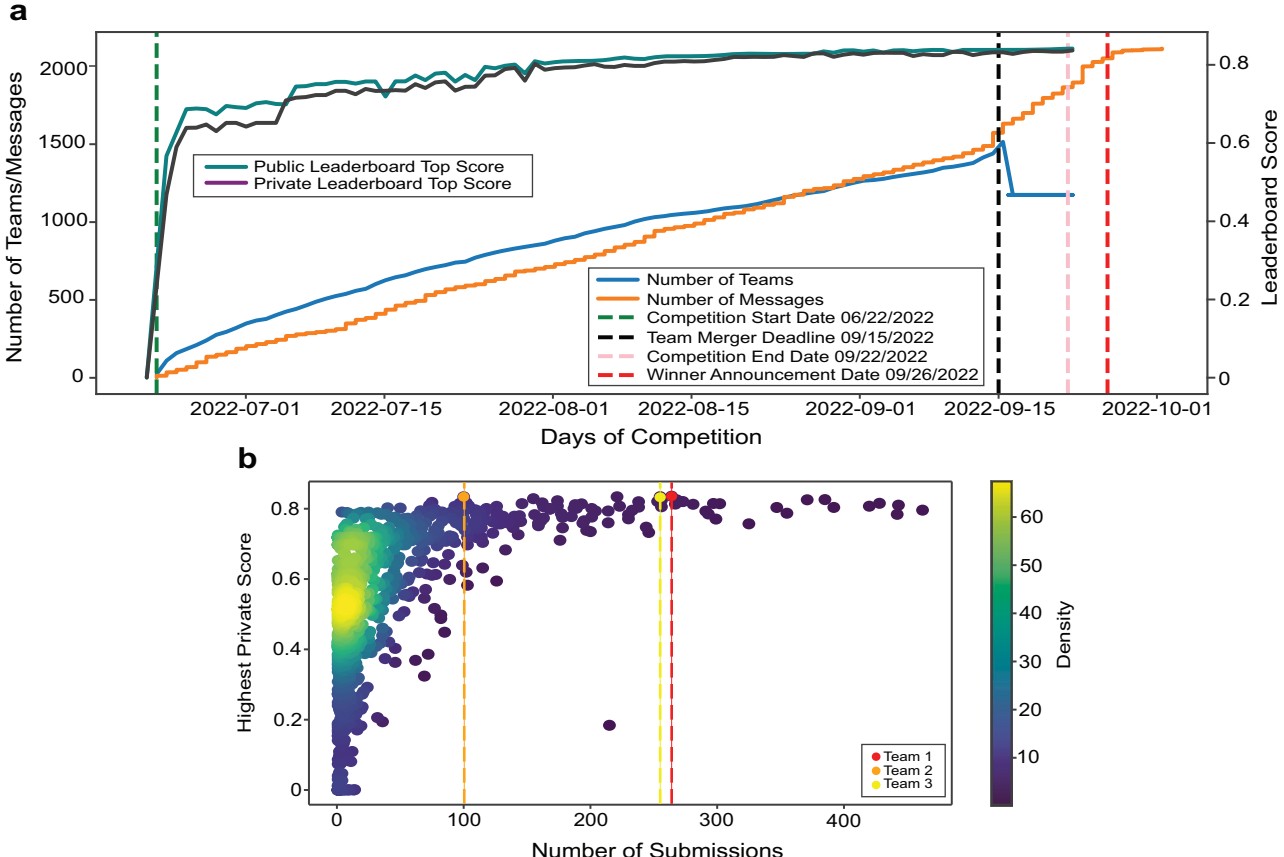

**Fig. 4 | Competition dynamics over 3 months. a** Number of teams and messages, and leaderboard high scores per day over competition period. **b** Number of submissions vs. highest private leaderboard score for each of the 1175 teams as a heatscatter. Source data are provided as a source data file on Zenodo (see "Data availability").

vision transformer models are much more sensitive to hyperparameter tuning and data changes than convolutional models; (3) model ensembles can be computationally expensive—especially during training—and might be impractical or inefficient for some production environments. Yet, this can be overcome either by using the single best model in the ensemble (at the cost of lower accuracy) or by employing techniques such as cascading[38] for faster inference.

Going forward, we plan to address the above-mentioned limitations by training and validating the models on more data, and optimizing the large ensembles for faster inference. The code from the winning models will be productized and deployed in the HuBMAP data portal to process large amounts of tissue data and extract biological knowledge in support of Human Reference Atlas construction and usage.

## Methods
### Competition and prizes
The "HuBMAP + HPA − Hacking the Human Body" competition was conducted on Google's ML and data science community platform called Kaggle, from June 22, 2022, to September 22, 2022, see Fig. 4a. The private leaderboard for identifying the three performance prize winners was finalized on September 26, 2022. The Judges' prize winners were announced 3 weeks later, after a thorough review and discussion by the judges' panel. The winners of the performance prize were awarded cash prizes (US$15,000 for first place; US$10,000 for second place; US$5,000 for third place). The winners of the Judges' prize were also awarded cash prizes (US$10,000 each for two scientific prizes; US$10,000 for one diversity prize).

**Performance prize.** A fast and efficient performance evaluation metric was required to score hundreds of submissions per day and a total of 39,568 submissions over three months. The teams submitted their inference code, after training their models, on the Submission portal. The submitted code was then run over the public test set to rank the teams on the public leaderboard. The teams typically use this score to validate their models. They can make an unlimited number of submissions before the competition deadline, but are limited to five submissions per day, see Fig. 4b. On competition end, the teams can choose up to two solutions to submit as their final submissions, which are then scored on the private test set (which remains inaccessible to the teams until winners are announced) to rank the teams on the private leaderboard. All scoring is done using the mean Dice score as the evaluation metric (see "Evaluation metrics" under "Methods") and the top-3 teams on the private leaderboard are selected as winners for the performance prize.

**Judges' prize.** Judges' prizes were aimed to promote experimentation, diversity, and science communication. The scientific prizes aimed to motivate solutions that go beyond the Dice evaluation metric and are more experimental in nature, providing insight into the dataset and/or computational techniques. The diversity and presentation prize promoted inclusion and the effective communication of scientific results. The winners were determined by a panel of human judges using a predefined and publicly available evaluation rubric (see Supplementary Note 5) that was publicly available on the Kaggle competition website at the competition start.

## Dataset collection and assembly

All tissue data used in this study is from donors examined and identified by pathologists as pathologically unremarkable tissue that can be used to derive the function of healthy cells. As the focus of this work is on the identification of FTUs, all images used in this competition feature at least one FTU.

**HPA data.** The HPA data consist of immunohistochemistry images of 1-mm-diameter tissue microarray cores and 4 μm thickness, stained with antibodies visualized with 3,3'-diaminobenzidine (DAB) and counter-stained hematoxylin (H)[19,39]. We retrieved over 7TB of public data from the HPA which comprised 23,610 images of 1mm diameter tissue microarray (TMA) cores for the large intestine, 27,906 for kidney, 28,098 for lung, 28,934 for prostate, and 27,474 for spleen. Given that the HRA aims to capture human adults, we removed all images associated with patients below the age of 18. We computed sex, age, tissue region percentage per image and selected 500 public images that maximize sex and age diversity per organ, have at least 1 FTU, and have a tissue region percentage above a threshold value (where threshold value is 5% for lung and 15% for kidney, spleen, large intestine, and prostate). The resulting dataset has 432 public HPA images distributed across the five organs (see Table 1). We further retrieved about 44GB of private data (not publicly available at the time of competition launch) from the HPA which comprised 295 images for kidney, 253 images for large intestine, 291 images for lung, 265 images for prostate, and 290 images for spleen. This dataset was processed in the same way as the public HPA data. A total of 81 images were selected for the final private dataset. All images, both public and private, are 3000 px × 3000 px (with some exceptions as roughly 19 images lie between 2308 px × 2308 px and 3070 px × 3070 px), and the diameter of each tissue area within an image is ~2500 px × 2500 px which corresponds to 1 mm. Hence, the pixel size of the images in this dataset is roughly 0.4 μm.

**HuBMAP data.** Multiple teams within or affiliated with HuBMAP shared 257 periodic acid-Schiff (PAS)[40] or hematoxylin and eosin (H&E)[41] stained WSIs of healthy human tissue that were not publicly available at the time of competition launch. From these WSIs, 1 mm × 1 mm tiles were extracted to match the size of the HPA TMA core images. Minimum donor metadata for all WSIs used in this competition included organ name, sex, and age. The resulting dataset had 448 image tiles distributed across the five organs and sourced from five different teams. All donors across all organs were above the age of 18, an exception being the spleen which included younger donors of ages 0 through 18. The pixel size of images across different organs was 0.5 μm for kidney, 0.229 μm for large intestine, 0.756 μm for lung, 6.263 μm for prostate, and 0.494 μm for spleen. The tissue slice thickness of all images in HuBMAP data was between 4–10 μm: 10 μm for kidney, 8 μm for large intestine, 5 μm for lung, 5 μm for prostate, and 4 μm for spleen.

**Dataset sampling.** Some images feature space without human tissue. We calculated the tissue region percentage for each image using Otsu's[42] thresholding. The specific threshold values for each organ were selected manually by analyzing the number of images available against different threshold values (see Supplementary Fig. 1). The values were selected such that images with very low actual tissue area are discarded, yet leaving a sufficient number of images to work with.

We then constructed a dataset with similar numbers of donor samples across age groups and sex for all organs (see Supplementary Fig. 2a)—insofar possible given available HuBMAP and HPA data. Note that systematic sampling of healthy human organ tissue is nontrivial; while human donors do not mind giving up adipose tissue, getting tissue from other organs is often only possible if an organ transplant cannot be executed or a patient dies, and the tissue is released for single-cell research. Consequently, the number of donors above the

age of 50 is higher than those below 50, especially for the HPA data (see Supplementary Fig. 2b).

**Data format.** For consistency, all images are exported as TIF files and all segmentations are provided as run-length encoded (RLE) masks for efficient storage and submissions (along with original JSON files) to the teams. Note that the RLE versions of the segmentation masks are cleaner than the JSON masks, although differences are minor. For example, the JSON versions might have segmentation overlaps that do not exist in the RLE copies but can also allow the teams to identify multiple adjacent FTUs, which would all end up in the same mask with RLEs.

## Acquiring ground truth labels and the final dataset

For four organs (except the kidney), 1–3 trained pathologists and/or anatomists (with experience in segmentation and histology) per organ provided initial segmentations done manually. For the kidney, the winning model from the previous HuBMAP Kaggle competition[4] was used to generate initial FTU segmentations for all HPA and HuBMAP kidney data, which were then manually reviewed and corrected by a professional anatomist.

All segmentations were verified and corrected through a final expert review process conducted by the lead pathologist for each organ. All images that were considered unsuitable were rejected. Partial FTUs were accepted, provided a human expert can segment it with confidence. All annotators, during the initial segmentation process as well as during the final review process, were given access to the images via an internal web-based segmentation tool (originally developed by the HPA team and further modified by the HuBMAP team). Please note that while extreme care was taken to get the best possible ground truth segmentations from experts, the labels do contain some noise, due to human bias, and existing issues were openly discussed on the public discussion forums of the competition.

**Final dataset.** The final dataset used in the competition contains 432 images from the HPA (including 351 public and 81 previously unpublished images with a total of 6,173 FTU annotations) and 448 previously unpublished images from HuBMAP (with a total of 6,728 FTU annotations) (see Fig. 1). All data are divided into three distinct datasets: a public training dataset containing all public HPA data (351 images), a public test dataset containing all previously unpublished HPA images (81 images) and HuBMAP images (209 images), and a private test dataset containing only HuBMAP images (239 images). The training dataset is openly accessible to the teams, while the test datasets remain hidden.

## Baseline segmentation model

To ensure the task is neither too easy (i.e., nearly 100% accuracy is achieved with little effort) nor too hard or impossible to accomplish (i.e., a satisfying accuracy is impossible), initial runs using the winning algorithm from the previous HuBMAP Kaggle competition, Tom, created a baseline model. The model was run on Indiana University's Carbonate large-memory compute cluster, using the GPU partition which consists of 24 Apollo 6500 GPU-accelerated nodes where each node is equipped with two Intel 6248 2.5 GHz 20-core CPUs. We used a single node with 300 GB of RAM and 2 Nvidia V100-PCIE-32GB GPUs.

The model required about 5 h for training and nearly 20 min for the inference task. It achieved a mean Dice score of 0.76 and 0.53 on the private HPA data and HuBMAP data, respectively. The mean Dice value achieved across the total private test dataset (HPA and HuBMAP) was 0.57 (see Supplementary Table 3). The same model achieved a mean Dice value of about 0.95 for the task of segmenting renal glomeruli in kidney images in the previous HuBMAP Kaggle competition. The results demonstrate the task is neither too easy nor too difficult, and there is a need for more generalizable algorithms.

## Evaluation metrics

The metric used to rank the performance of the teams in the competition is mean Dice coefficient[22,43] (also referred to as the mean Dice score). The Dice score compares the pixel-wise agreement between a predicted segmentation (PS) and its corresponding ground truth segmentation (GT) for an image: $\frac{2*|GT \cap PS|}{|GT| + |PS|}$.

The leaderboard score used is the mean of the Dice coefficients for each image in the test set. It should be noted that calculation of Dice coefficient does not take into account separation between individual instances. Hence, in case multiple predicted FTUs overlap/merge, the Dice coefficient for that prediction may still be high while the FTU count may be incorrect (and might require further processing, either programmatic or manual, to separate the individual instances of FTUs).

After extensive discussion of options with the Kaggle data scientists and machine learning experts from the panel of judges, the mean Dice coefficient was selected for performance prize ranking. While other metrics such as the mean Average Precision[44] (mAP) might have been better suited for the problem, the Kaggle team recommended going forward with the mean Dice score, taking into account the nature of the dataset and timeline for the competition. Dice is a well-tested metric used in many competitions on the Kaggle platform and other metrics require much more testing by the Kaggle team to ensure participants cannot find loopholes and exploit vulnerabilities in the metric during the competition. Hence, while Dice score may not be the ideal metric[45,46] in a production setting, it is a good enough metric to evaluate and compare solutions from Kaggle competitions.

The post-hoc analysis uses mean Intersection-Over-Union (IOU) (also known as Jaccard index[32,33]) as an auxiliary metric to further test the predictions and rankings. The IOU is defined by $IOU(A,B) = \frac{|A \cap B|}{|A \cup B|} = \frac{|A \cap B|}{|A| + |B| - |A \cap B|}$, where A and B are the two objects being compared (e.g., GT and PS). It represents the proportion of area of overlap out of the area of union for the two objects. The Dice coefficient and the IOU are always within a factor of 2 of each other, and while they are generally positively correlated—especially for individual images—differences may emerge when taking the mean over a dataset. The IOU tends to penalize incorrect predictions more, quantitatively, and hence has a squaring effect on the errors. While Dice measures average performance, the IOU measures worst-case performance.

## Public and private leaderboards

Kaggle ranks teams on two leaderboards—public leaderboard and private leaderboard—each using a different subset of the test data, using the predetermined evaluation metric for the performance prize. The public leaderboard uses the public test data, and the private leaderboard uses the private test data. The public leaderboard rankings and scores are visible to the teams and are used to validate their algorithms, providing feedback they can use to improve their algorithms. The private leaderboard rankings remain hidden to the teams until the end of the competition to ensure algorithms are not overfitted to the test data. The top-3 teams on the private leaderboard are considered as winners of the performance prizes.

## Participation analysis

At the conclusion of the competition, participation metadata becomes publicly available on Meta Kaggle[47]—Kaggle's public data on competitions, users, submission scores and kernels. Meta Kaggle tables were initiated in 2015 and are updated daily with information on completed competitions. We use these data to understand how the Hacking the Human Body competition unfolded over its 3-month period.

We use standard python packages for data science such as Pandas[48], NumPy[49], Matplotlib[50], and Seaborn[51] for running all analyses; creating all visualizations in Jupyter[52] Notebooks. The analyses can be replicated for any competition on Meta Kaggle using the code we made available on GitHub (see "Code availability").

## Statistical analysis

To assess the impact of worst-case predictions on the rankings of the three winning teams relative to each other, a modified leave-one-out[53] analysis is conducted and evaluated with both a Dice score and an IOU score. When removing the worst five cases from each team per organ (25 cases in total), the rankings remain the same with mean Dice scores of 0.8463, 0.8452, and 0.8441 and mean IOU scores of 0.7497, 0.7480, and 0.7451. Leaving out one worst case for each organ (five cases in total), the rankings stay the same but leads to a very small difference between the scores for the three teams (mean Dice scores of 0.8421, 0.8418, and 0.8413, and mean IOU scores of 0.7452, 0.7446, and 0.7423, respectively). Finally, leaving out three worst cases for each organ (15 cases in total), team 3 ranks first based on the mean Dice score (0.8505, 0.8503, and 0.8508), but the rankings based on mean IOU stay the same (0.7554, 0.7548, and 0.7538).

The ranking stability for the top-50 teams is further assessed by calculating Kendall's Tau[53–55] (also called Kendall's Rank Correlation) which is used to quantify the agreement between two rankings and is independent of the number of entities ranked. Tau values closer to 1 show a strong positive correlation between the two rankings, where a value of 1 would mean perfect alignment. A p-value associated with the tau value indicates the statistical significance of the correlation. Lower p-values (closer to 0) indicate higher significance of the relationship between the two rankings such that it is unlikely to occur by chance. The tau between ranking for the top-50 teams based on mean Dice score and mean IOU score is 0.74 (p-value = 1.9505e-14). If the worst case per organ is dropped for each team, the tau is 0.75 (p-value = 1.5026e-14) and if the worst three cases per organ are dropped, the tau is 0.73 (p-value = 7.0708e-14). In addition, if the competition would have been ranked based on mean IOU, instead of mean Dice, while the top-50 rankings changed to some extent, the three winning teams rank the same. The mean Dice score and the mean IOU score for the top-50 teams is provided in Supplementary Table 4. Kendall's tau is computed using the implementation in the Python Scipy[56] library.

## Statistics and reproducibility

The final dataset used for the competition was curated from a larger pool of data available from HuBMAP and HPA. The images were selected such that a balance can be maintained across sex and age groups. Images were selected from both HuBMAP and HPA data such as to maintain balance between both sources. Images with damaged or unhealthy tissue were excluded, and images containing very low tissue region percentage were also excluded for the final dataset. All code submissions for inference were collected and graded automatically, which allows for the reproduction of the scores. The final ranking was determined after re-running all team's chosen submissions. On the competition end, all winning algorithms were validated and compared to scores on competition leaderboard. This was done once to validate the results and grant prizes to the top-performing teams. The assignment of data sources to train/test sets was intentional to maintain specific data sources across train/test sets but the assignment of individual images in the specific train/test sets was random. The only criteria used was to balance donor sex and age across these datasets. Since the primary purpose of this final dataset was to build machine learning models, the randomization provides the advantage of the algorithms not overfitting to human bias during sampling. The pathology experts that annotated the ground truth were aware of the specific tissue they were annotating but not necessarily aware of the donor metadata associated with it. The teams in the competition had access to the metadata associated with the public training data but did not have access to any information regarding the private test set (which was used for competition ranking and deciding winners).

## Reporting summary

Further information on research design is available in the Nature Portfolio Reporting Summary linked to this article.

## Data availability

All curated data used in the competition (HuBMAP and HPA), along with the trained models from the winning teams, is publicly available via Zenodo. All primary competition data, as well as external data used by Teams 1 and 2, is published on Zenodo at https://doi.org/10.5281/zenodo.7545744. All external data used by Team 3 is available as Kaggle datasets, links to which are provided in the Supplementary Information. All trained model weights are published on Zenodo at https://doi.org/10.5281/zenodo.7545792. Source data for all plots presented in the paper are provided as a Zenodo dataset at https://doi.org/10.5281/zenodo.8144891.

## Code availability

All code used for data preprocessing and analysis, baseline model, winning algorithms, and participant analysis are publicly available on GitHub https://github.com/cns-iu/ccf-research-kaggle-2022. An archived version of this code is also published on Zenodo and is made publicly available (https://doi.org/10.5281/zenodo.8144891).

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

## Acknowledgements

We appreciate the generosity of the sponsors of this competition: Google HCLS and Genentech (a member of the Roche group). We thank Andrea de Souza (Eli Lilly and Company) for obtaining sponsors. We thank Amy Kemper, Sohier Dane, and Addison Howard (Google/Kaggle team) for expert support throughout the competition. We thank the HPA personnel Mattias Forsberg and Kalle von Feilitzen (both from Royal Institute of Technology) for assistance in making the images accessible to the challenge and HuBMAP teams for providing data and segmentation expertise for five organs: Jeff Spraggins (VU), John Hickey (Stanford University), Gloria Pryhuber (URMC), Doug Strand (UTSouthwestern), Maigan Brusko (UFL), Sanjay Jain (Washington University School of Medicine in St. Louis), Jeanne Shen (Stanford University), Iain Miller (Stanford University), Benjamin Dulken (Stanford University), Gail Deutsch (University of Washington). We would also like to thank Andrew Hull (CU Anschutz), Alexis Macdonald (CU Anschutz), Monica Fong (CU Anschutz), and Hinrich Freitag (Hannover Medical School) for providing their time and expertise to manually segment the datasets. Mike Gallant (Indiana University) and Jason Swedlow (University of Dundee) kindly provided assistance for transferring and compiling the HPA data. Bruce W. Herr II (Indiana University) helped provision the web-based segmentation tool. Naveksha Sood (Indiana University) helped run code for generating pre-segmentations on HPA and HuBMAP data. We appreciate the work done by Rachel Bajema (Indiana University) on the illustrations and figures presented in the paper. Organ illustrations in Fig. 1 created and provided by Leonard Cross and Heidi Schlehlein (Indiana University). We are grateful to the Kaggle Scientific and Diversity prize judges Zorina Galis (NIH), Carolina Wählby (Uppsala University), Artem Sokolov (HMS), Constantin Kappel (Leica Microsystems), Anna Kreshuk (EMBL), Blue Lake (UC San Diego), David van Valen (CalTech), Jhimli Mitra (GE Research), Nathan Heath Patterson (VU), and Bobak Kechavarzi (Cleveland Clinic) for sharing their time and expertise. This research has been funded in part by the NIH Common Fund through the Office of Strategic Coordination/Office of the NIH Director under award OT2OD026671 (K.B.) and OT2OD033756 (K.B.), NIH awards U54EY032442-01 (K.B.) and U54HG010426-01 (K.B.), National Institute of Diabetes and Digestive and Kidney Diseases (NIDDK) award U54DK120058 (K.B.), the Kidney Precision Medicine Project grant U2CDK114886 (K.B.), and the Knut and Alice Wallenberg Foundation (E.L. and C.L.). The content is solely the responsibility of the authors and does not necessarily represent the official views of the National Institutes of Health.

## Author contributions

Y.J. aided the design and implementation of the competition; led the data analysis, sampling and processing; refactored and maintained a web-based segmentation tool; aided segmentation and review process by experts; oversaw the competition throughout its duration; acted as liaison to data providers, segmentation experts, HPA team and Kaggle Team; co-wrote the paper. L.L.G. aided the design and implementation of the competition; led data and metadata collection from HuBMAP data providers; aided data segmentation and review process by experts; acted as liaison to data providers, segmentation experts, HPA team and Kaggle Team; implemented initial code for participation analysis and visualizations; co-wrote the paper. S.J. contributed to data analysis, sampling, and processing; conducted baseline model training and analysis. S.M. ran participation analysis and rendered the resulting visualizations. T.L. aided implementation of the competition; helped using the web-based segmentation tool; performed data and metadata assembly from HPA. C.L. aided design and implementation of the competition; guided data and metadata assembly from HPA; led the pathology-based data generation of the original HPA dataset. E.L. aided the design and implementation of the competition; led data and metadata assembly from HPA. K.B. led the design and implementation of the competition; co-wrote the paper.

## Competing interests

The authors declare no competing interests.
