## [Peer Review File · Nature Communications]

Reviewers' Comments:

Reviewer #1:

None

Reviewer #2:

Remarks to the Author:

First, I would like to thank the authors for their comprehensive response and addressing some of the raised concerns. Especially, the added background on FTUs and the provided reasoning for the scientific and diversity prizes were addressed.

While the additional information regarding the selection of the Dice metric as the primary evaluation metric is helpful and relatable, the previously raised problems of the chosen metric remain. Considering that mAP is the most predominant evaluation metric in the object detection domain and is actively used for many years by several large benchmarks which shows the broad adoption of the metric. Furthermore, other Kaggle Challenges also utilized metrics to quantify the detection performance and thus it remains unclear if there might have been a better solution for this. While this can not be changed after concluding the challenge, some kind of discussion or qualitative post-hoc analysis could have been provided in the manuscript.

Nevertheless, since semantic segmentation was chosen as the task formulation of the challenge, an extended analysis in the provided manuscript should be added to shed some light on dis-/advantages of the top performing solutions. Some examples are listed below:

- * analysis of individual failure cases, i.e. did all the algorithms fail at the same FTUs? Were there individual images which had particularly bad performance or were only single FTUs missed?
- * additional visualizations of the final results in terms of violin- and/or bar plots could be used to provide additional evidence on the previously mentioned point.
- * [optional, since this is probably less relevant in the context of FTU segmentation] boundary based metrics such as the boundary dice could be used to analyze the behavior of the algorithms at the boundaries of the FTUs and potentially provide additional insights into algorithmic performance.

Without the above mentioned points the current manuscript lacks methodological insight for future competition organizers or participants. Furthermore, post hoc analysis on the stability of the selected challenge metric (and potentially additional auxiliary metrics) could be conducted to investigate the stability of the ranking. All of these should be backed by some commonly used statistical tests.

While the provided ablation experiments in the appendix already provide a small glimpse of the top performing algorithms, much more detailed information on the employed training strategies should be added to the manuscript (potentially to the supplement), some examples are given below:

- * Detailed information on the used ensembles for each of the three top performing methods (while the text in the main body gives a rough outline of the used model, more detailed in the supplement would be highly appreciated, e.g. which conv nets were used inside the ensembles)
- * Which external data sets were used by the top performing methods? This is important information since external data could be one of the driving factors of winning solutions and employing more or different external data might be as important as choosing the right model
- * Did all teams use the same pseudo labeling techniques or were there differences?

- * Ablation experiments of Team 2 are missing and could potentially provide some additional insight into their experiments.
- * The tables in the appendix are very hard to comprehend and thus require more detailed explanations of the experiments and changes.
- * Qualitative results from the predictions of the methods should be added to highlight success and failure cases of the methods.
- * [optional] Due to the large number of teams, the detailed analysis could be extended to the top 5 or even 10 performing teams in order to provide a more comprehensive insight of the methods.

In conclusion, the challenge attracted a lot of interest by the community and a very large number of participants competed in the presented challenge highlighting its impact on the domain and the general interest in FTU segmentation. Unfortunately, the current descriptions of the top performing methods are not detailed enough and an extended evaluation is needed to fully leverage the available information. In its current form, the manuscript is limited to rather shallow methodological insight from the top performing methods.

Point-by-Point Response to Reviewer Comments

Reviewer #2 (Remarks to the Author):

First, I would like to thank the authors for their comprehensive response and addressing some of the raised concerns. Especially, the added background on FTUs and the provided reasoning for the scientific and diversity prizes were addressed. While the additional information regarding the selection of the Dice metric as the primary evaluation metric is helpful and relatable, the previously raised problems of the chosen metric remain. Considering that mAP is the most predominant evaluation metric in the object detection domain and is actively used for many years by several large benchmarks which shows the broad adoption of the metric. Furthermore, other Kaggle Challenges also utilized metrics to quantify the detection performance and thus it remains unclear if there might have been a better solution for this. While this can not be changed after concluding the challenge, some kind of discussion or qualitative post-hoc analysis could have been provided in the manuscript.

Authors: Thank you for your expert comments. We have now added further discussion and a post-hoc analysis to the manuscript (see **Qualitative Analysis of Predictions** in **Results** and **Statistical Analysis** in **Methods**). Specifically, we have added Intersection-Over-Union (IOU)--another popular metric for semantic segmentation tasks--as an auxiliary metric and computed that for the top-3 winning teams, as well as the top-50 teams, to assess how a different metric would have impacted the results. We found that while using IOU leads to some changes in the top-50 team rankings, the top-3 teams rank the same.

Nevertheless, since semantic segmentation was chosen as the task formulation of the challenge, an extended analysis in the provided manuscript should be added to shed some light on dis-/advantages of the top performing solutions. Some examples are listed below:
* analysis of individual failure cases, i.e. did all the algorithms fail at the same FTUs? Were there individual images which had particularly bad performance or were only single FTUs missed?

Authors: We have now added qualitative results on the top five and worst five cases for each organ for all three winning solutions. The findings have been added to the **Qualitative Analysis of Predictions** section under **Results**. Additionally, we now provide a visual comparison of per-pixel false positives and false negatives for these 10 cases times 3 methods times 5 organs in the **Supplementary Information**.

*** additional visualizations of the final results in terms of violin- and/or bar plots could be used to provide additional evidence on the previously mentioned point.**

Authors: We have now added a Figure 4 which shows the violin plots for mean Dice scores and mean IOU scores for all 3 winning teams, broken down by organs. For each violin plot, the individual image scores are also plotted as a swarm plot overlaid on top of the violin plots to highlight the spread and outliers.

*** [optional, since this is probably less relevant in the context of FTU segmentation] boundary-based metrics such as the boundary dice could be used to analyze the behavior of the algorithms at the boundaries of the FTUs and potentially provide additional insights into algorithmic performance.**

Authors: Considering the varying number of instances per image and the presence of touching FTUs, we decided not to compute the boundary based metrics such as Hausdorff Distance and Hausdorff Distance at 95th percentile. We made this choice based on the “Metrics Reloaded” paper (<https://arxiv.org/abs/2206.01653>) and the detailed rubrics provided within.

Without the above mentioned points the current manuscript lacks methodological insight for future competition organizers or participants. Furthermore, post hoc analysis on the stability of the selected challenge metric (and potentially additional auxiliary metrics) could be conducted to investigate the stability of the ranking. All of these should be backed by some commonly used statistical tests.

Authors: We have added mean IOU scores for the top-50 teams and compared how that would affect the rankings. We found that while using IOU leads to some changes in the top-50 team rankings, the top-3 teams rank the same. Additionally, we have added a study detailing how removing the worst predictions impacts the scores and the rankings. While the scores improve slightly, the rankings stay the same in all cases, except when removing the worst three cases per organ, team 3 ranks first based on dice score. We computed Kendall's Rank Correlation to further investigate the ranking stability and observe high correlation between the rankings based on mean dice score and mean IOU score but not a perfect alignment. These results have been added to the **Statistical Analysis** section under **Methods**.

While the provided ablation experiments in the appendix already provide a small glimpse of the top performing algorithms, much more detailed information on the employed training strategies should be added to the manuscript (potentially to the supplement), some examples are given below:

*** Detailed information on the used ensembles for each of the three top performing methods (while the text in the main body gives a rough outline of the used model, more detailed in the supplement would be highly appreciated, e.g. which conv nets were used inside the ensembles)**

*** Which external data sets were used by the top performing methods? This is important information since external data could be one of the driving factors of winning solutions and employing more or different external data might be as important as choosing the right model**

*** Did all teams use the same pseudo labeling techniques or were there differences?**

Authors: We have added detailed information on the model architectures, training details, external data, and pseudo labeling techniques of the three winning teams to the Supplementary Information.

*** Ablation experiments of Team 2 are missing and could potentially provide some additional insight into their experiments.**

Authors: Unfortunately, since ablation study is not a part of the final submission, not all teams track their experiments. While team 1 and team 3 provided their experiments voluntarily, team 2 did not. We reached out to Team 2 and they informed us that they do not have this information.

*** The tables in the appendix are very hard to comprehend and thus require more detailed explanations of the experiments and changes.**

Authors: We have now added a further explanation of the ablation studies to the tables in the supplementary information.

*** Qualitative results from the predictions of the methods should be added to highlight success and failure cases of the methods.**

Authors: We have now added figures of the five best and five worst predictions per organ for all three winning teams to the supplementary information.

*** [optional] Due to the large number of teams, the detailed analysis could be extended to the top 5 or even 10 performing teams in order to provide a more comprehensive insight of the methods.**

Authors: Since only the winning teams are required to submit a detailed documentation as well as training code of their solutions, most teams that do not win don't provide this information. While some teams may choose to post some information in the Discussion forums on the Kaggle competition website, it is generally not very thorough.

In conclusion, the challenge attracted a lot of interest by the community and a very large number of participants competed in the presented challenge highlighting its impact on the domain and the general interest in FTU segmentation. Unfortunately, the current descriptions of the top performing methods are not detailed enough and an extended evaluation is needed to fully leverage the available information. In its current form, the manuscript is limited to rather shallow methodological insight from the top performing methods.

We hope that the proposed changes in the current manuscript address your concerns about this work.

Reviewers' Comments:

Reviewer #2:

Remarks to the Author:

I would like to thank the authors for the revised manuscript and incorporating the previously mentioned feedback. The newly added information in the form of qualitative results, violin plots about the score distributions and extended training descriptions of the best three performing teams adds a lot of value to the provided manuscript.

Several open points remain after revision of the manuscript:

Improved scientific nomenclature should be used when reporting information about the p-value, e.g. by specifying the exact value or indicating if it is below/below commonly used significance thresholds, like $p < 0.05$ or $p < 0.01$. The manuscript currently only mentions "p-value tends to 0", which is not helpful for the reader to get an idea of the stability of the results. Maybe it would be easier for the authors to use out-of-the-box frameworks like <https://github.com/wiesenfachallengeR> to create a more comprehensive evaluation of the results.

The authors extended the initial evaluation with an additional semantic segmentation metric, namely "mean IoU", to improve the provided analysis. As the manuscript already points out correctly, the Dice coefficient and the IoU are directly connected to each other and thus measure the same properties of the provided algorithms. As such the added value by using a related metric to the manuscript remains questionable (this is also pointed out in Reinke, Annika, et al. "Understanding metric-related pitfalls in image analysis validation." ArXiv (2023). - Figure 39). Currently, the Dice score is computed as the average across all images without considering the different organs. A more sophisticated aggregation scheme to account for the different classes might have been a better solution (even though the ranking of the top performing methods remains the same). Even though the overall ranking remains the same, this nevertheless leads to new perspectives on the methods (e.g., 0.003 vs. 0.008 for 1st rank vs. 3rd rank).

Considering that object level evaluation was used in previous challenges (also on Kaggle), e.g. Caicedo, Juan C., et al. "Nucleus segmentation across imaging experiments: the 2018 Data Science Bowl." Nature methods 16.12 (2019): 1247-1253. and semantic segmentation is not able to fully grasp the targeted requirements; the evaluation remains the biggest weakness of the manuscript.

The manuscript was significantly improved during the revision and now contains insightful information about the submitted methods. The additional analysis with the "mean IoU" metric adds a second metric but additional insights are limited since the original Dice metric measured the same properties. The evaluation of methods remains the primary weakness of the manuscript.

Reviewer #2

I would like to thank the authors for the revised manuscript and incorporating the previously mentioned feedback. The newly added information in the form of qualitative results, violin plots about the score distributions and extended training descriptions of the best three performing teams adds a lot of value to the provided manuscript.

Authors: Thank you again for your constructive feedback. Glad to hear that our revisions meet your expectations.

Several open points remain after revision of the manuscript:

Improved scientific nomenclature should be used when reporting information about the p-value, e.g. by specifying the exact value or indicating if it is below/below commonly used significance thresholds, like $p < 0.05$ or $p < 0.01$. The manuscript currently only mentions “p-value tends to 0”, which is not helpful for the reader to get an idea of the stability of the results. Maybe it would be easier for the authors to use out-of-the-box frameworks like <https://github.com/wiesenfa/challengeR> to create a more comprehensive evaluation of the results.

Authors: Thank you for suggesting. We have revised this in the final manuscript.

The authors extended the initial evaluation with an additional semantic segmentation metric, namely “mean IoU”, to improve the provided analysis. As the manuscript already points out correctly, the Dice coefficient and the IoU are directly connected to each other and thus measure the same properties of the provided algorithms. As such the added value by using a related metric to the manuscript remains questionable (this is also pointed out in Reinke, Annika, et al. "Understanding metric-related pitfalls in image analysis validation." ArXiv (2023). - Figure 39).

Authors: While both IOU and Dice coefficient measure the same properties, we decided to add IOU to check for deviations when averages are computed (as pointed out in Maier-Hein, Lena, and Bjoern Menze. "Metrics reloaded: Pitfalls and recommendations for image analysis validation." *arXiv.org* 2206.01653 (2022)). Additionally, we also added IOU to compare the effects of this metric on competition rankings, since IOU is also a major metric used in semantic segmentation tasks. Results show that while the winning teams stay the same, some differences are present in team rankings for top-50 based on this metric. While boundary based metrics would

have provided more insight into the methods, due to the presence of overlapping/touching instances, we decided not to use such metrics, as discussed in the manuscript.

Currently, the Dice score is computed as the average across all images without considering the different organs. A more sophisticated aggregation scheme to account for the different classes might have been a better solution (even though the ranking of the top performing methods remains the same). Even though the overall ranking remains the same, this nevertheless leads to new perspectives on the methods (e.g., 0.003 vs. 0.008 for 1st rank vs. 3rd rank).

Authors: We now provide the mean dice scores per organ for the top-3 teams in Table 2. This gives more insight into the methods used by the three winning teams in terms of their performance per organ. Extending this beyond the winning teams does not seem valuable, since the methods used by those teams are not available.

Considering that object level evaluation was used in previous challenges (also on Kaggle), e.g. Caicedo, Juan C., et al. "Nucleus segmentation across imaging experiments: the 2018 Data Science Bowl." Nature methods 16.12 (2019): 1247-1253. and semantic segmentation is not able to fully grasp the targeted requirements; the evaluation remains the biggest weakness of the manuscript.

The manuscript was significantly improved during the revision and now contains insightful information about the submitted methods. The additional analysis with the "mean IoU" metric adds a second metric but additional insights are limited since the original Dice metric measured the same properties. The evaluation of methods remains the primary weakness of the manuscript.

Authors: While we agree that instance segmentation might have been a better fit for the problem at hand, posing the problem as a semantic segmentation problem does not limit the utility of the work. The main insights into methods from the participating and winning teams comes from their handling of variability in the datasets. The methods developed by the teams to tackle variability in tissue staining, image resolutions, FTU shapes and sizes and structures, are the main contributions of the challenge. If the competition was posed as an instance segmentation problem, these insights would have likely remained the same. Since the development of the Human Reference Atlas is concerned with populations instead of individuals, segmenting FTU regions in images is of great value for other downstream analyses, even without differentiating between instances.